# Protective Role of Hepassocin against Hepatic Endoplasmic Reticulum Stress in Mice

**DOI:** 10.3390/ijms232113325

**Published:** 2022-11-01

**Authors:** Yang Yang, Hui Chen, Yue Wan, Diandian Dong, Xiaofang Wang, Songhui Yao, Pengjun Wang, Shensi Xiang, Xiaoming Yang, Miao Yu

**Affiliations:** 1Department of Pharmaceutical Engineering, School of Chemical Engineering and Technology, Tianjin University, Tianjin 300072, China; 2Beijing Institute of Radiation Medicine, Beijing 100850, China; 3School of Basic Medical Sciences, An Hui Medical University, Hefei 230032, China; 4Institute of Life Sciences, He Bei University, Baoding 071002, China; 5State Key Laboratory of Proteomics, Beijing Proteome Research Center, National Center for Protein Sciences (Beijing), Institute of Lifeomics, Beijing 102206, China

**Keywords:** hepassocin, ER stress, lipid accumulation, liver injury

## Abstract

Hepassocin (HPS) is a hepatokine that has multiple proposed physiological functions. Some of the biological processes in which it is involved are closely related to endoplasmic reticulum (ER) stress, but the role of HPS in the regulation of ER stress remains unclear. Here, we demonstrated that *HPS* transcription is induced by the protein kinase RNA-like ER kinase (PERK)/activating transcription factor 4 (ATF4) cascade upon ER stress in hepatocytes. Additionally, fasting/refeeding also induced HPS expression in mice liver. The loss of HPS sensitizes hepatocytes to ER stress-related cytotoxicity in vitro, whereas HPS treatment altered these phenotypes. HPS deficiency exacerbates fasting/refeeding-induced ER stress in vivo. The preliminary administration of HPS ameliorates liver steatosis, cell death, and inflammation in mice injected with tunicamycin (TM). The improvement of HPS can be observed even if HPS protein is injected after TM treatment. Furthermore, the administration of an ER stress inhibitor alleviated steatohepatitis in methionine- and choline-deficient (MCD) diet-fed HPS-deficient mice. These results suggest that HPS protects hepatocytes from physiological and pathological ER stress, and that the inactivation of HPS signaling aggravating ER stress may be a novel mechanism that drives the development of steatohepatitis. The protective mechanism of HPS against ER stress in hepatocytes was associated with the regulation of ER calcium handling, and the suppression of calcium influx release from ER upon stressor treatment. Collectively, our findings indicate that HPS may act in a negative feedback fashion to regulate hepatic ER stress and protect hepatocytes from ER stress-related injury. HPS has the potential to be a candidate drug for the treatment of ER stress-related liver injury.

## 1. Introduction

As an important organelle, the endoplasmic reticulum (ER) is responsible for various biological processes such as protein folding, lipid synthesis, calcium storage, and signal transduction in cells. In some physiological, pathological and pharmacological situations, ER dysfunction leads to a decreased protein-folding capacity, which triggers the unfolded protein response (UPR). When the levels of unfolded and misfolded proteins in the ER are out of control, ER stress is activated [1,2]. Accumulating evidence has revealed that hepatic ER stress is initiated by physiological [3] and pathological insults [4,5]. Excessive and/or sustained ER stress contributes to a variety of liver diseases, such as drug-induced liver failure [6], hepatitis C [7], cholestatic syndrome [8] and nonalcoholic fatty liver (NAFLD) [9]. Abnormal ER stress is therefore considered as a cardinal feature of liver injury. A better understanding of the related molecular mechanisms is of critical importance for clinical prognostication and the development of drug candidates for liver disease.

Hepassocin (HPS) is mainly expressed and secreted by hepatocytes, and it promotes cell proliferation in an autocrine manner [10,11,12]. Moreover, it also shows a low level of expression in adipose tissue and the pancreas [13]. HPS is considered as a key metabolism regulator, since HPS-deficient mice exhibit a global metabolic defect phenotype [13]. Further studies demonstrated that HPS is related to the development of fatty liver disease and obesity [14,15,16]. Recently, HPS has been shown to be a functional ligand of the lymphocyte-activation gene 3 (LAG-3), inhibiting T-cell activation, and the ablation of HPS in aged mice causes autoimmune disease [17]. Our and other previous publications have shown that HPS plays a key protective role in chemotoxicity-, glucotoxicity-, and lipid toxicity-induced liver injury [16,18,19,20]. Interestingly, HPS is induced in mice hepatocytes or liver by ER stress inducers such as tunicamycin (TM) and palmitic acid (PA) [20,21]. We recently reported that HPS deficiency exacerbates ER stress responses in PA-treated mice hepatocytes or liver with nonalcoholic steatohepatitis (NASH) [16]. Moreover, the administration of recombinant HPS protein inhibited PA-induced ER stress in hepatocytes [16]. This highlights the potential role of HPS in ameliorating hepatocyte injury by modulating ER stress. However, the roles and molecular mechanisms of HPS in the regulation of ER stress remain largely undefined.

Here, our in vivo and in vitro evaluations demonstrate that HPS may regulate hepatic ER stress in a negative feedback manner under physiological as well as drug-induced conditions. Additionally, we show that the inhibition of ER stress may be an important mechanism for HPS in the attenuation of NASH development in mice. Our results imply that HPS or its signaling pathway might have therapeutic potential for ER stress-related liver injury.

## 2. Results

### 2.1. ER stress Induces HPS Expression in Hepatocytes through PERK-ATF4 Pathway

We confirmed and extended previous findings [20,21] through the treatment of HepG2 cells or mice with an ER stress inducer—tunicamycin (TM). The results showed that TM gradually and significantly increased *HPS* mRNA and protein levels in HepG2 cells (Figure 1A,B). Similarly, the ability of TM to induce *HPS* mRNA and protein levels was proven in mice liver, as well as to elevate the plasma levels of HPS (Figure 1C–E). Moreover, TM treatment significantly increased the luciferase activities of human *HPS* promoter, which is contained ~2.5 kb upstream of the *HPS* transcription start site [22] in HepG2 cells (Figure 1F). Similar effects were also observed in mice transfected with HPS-luc reporter using a hydrodynamic method (Figure 1G). These data suggest that ER stress might induce *HPS* expression at the transcriptional level. Through pharmacological inhibition and assessing TM-induced HPS protein expression, we observed the pharmacological inhibition of inositol-requiring enzyme 1 (IRE1) by the compound STF-083010, which can inhibit the endonuclease activity of IRE1 [23], as evidenced by decreased XBP1s protein expression having no effect on TM-induced HPS expression. In contrast, the PERK inhibitor GSK-2656157 [24], which can suppress the phosphorylation of PERK, significantly blunted TM-induced HPS expression in HepG2 cells (Figure 1H), suggesting that the PERK cascade in the ER stress response regulates HPS expression in hepatocytes. The effect of TM on HPS-luciferase reporter was mimicked by the overexpression of ATF4, the downstream transcription factor of PERK, even in the absence of ER stress inducers in HepG2 cells (Figure 1I). We analyzed the promoter structure of *HPS* using PSCAN online software (http://159.149.160.88/pscan/ (accessed on 4 June 2021)), and found a putative binding site for CEBPB (5′-GTTACACAAT-3′) -635bp upstream of the *HPS* transcription start site, which is a target of ATF4 and has been shown to contribute to *HPS* expression in ER stress [21]. We demonstrated that CEBPB was significantly increased following TM treatment (Figure 1B), and the deletion of the binding site for CEBPB did not affect the basal transcriptional activity of the *HPS* promoter but significantly reduced the response to the overexpression of ATF4 (Figure 1I).

### 2.2. HPS Is Responsible for ER Stress Protection in Hepatocytes In Vitro

HPS-KO hepatocytes showed a significant decrease in resistance toward cytotoxicity induced by thapsigargin (TG), an ER stressor [25], as shown by the decreased cell survival (Figure 2A) and elevated levels of LDH release (Figure 2B) and apoptosis rate (Figure 2C). HPS treatment rescued the ER stress-sensitive phenotypes of the HPS-KO hepatocytes (Figure 2A–C), confirming the protective role of HPS in TG-induced cytotoxicity. HPS-KO hepatocytes also had an increased apoptosis rate compared to WT hepatocytes following TM treatment, whereas HPS treatment prevented increased apoptosis in TM-treated HPS-KO hepatocytes (Figure 2D). For HPS-KO and WT hepatocytes, TM treatment increased pro-apoptotic Bax and activated Caspase-9 expression (Figure 2E). These data verify the role of HPS in the prevention of hepatic apoptosis caused by ER stress. In keeping with these results, either TM or TG treatment increased the protein levels of ER chaperone BiP (GRP78) and the phosphorylation of eIF2α, the ER stress marker, in HPS-KO hepatocytes compared with the WT hepatocytes (Figure 2F). Accordingly, HPS treatment inhibited the increase in mRNA expressions of ER stress response genes, including *Grp78*, *DNA-damage inducible transcript 3* (*Ddit3*/*Chop*), *derlin-3* (*Derl3*), *Synoviolin 1* (*Syvn1*/*Hrd1*), *DnaJ heat shock protein family (Hsp40) member B9* (*Dnajb9*/*Erdj4*), and *Homocysteine-inducible, endoplasmic reticulum stress-inducible, ubiquitin-like domain member 1* (*Herpud1*) in a dose-dependent manner (Figure 2G). HPS treatment also decreased GRP78 protein, eIF2α and PERK phosphorylation in the WT hepatocytes upon TM challenge (Figure 2H). These results suggest that HPS may directly counteract ER stress in hepatocytes. Notably, HPS did not affect the TM-induced upregulation of ER stress response genes in non-hepatocyte-derived cells, including HEK293 and HeLa cells (Appendix A), suggesting that HPS’s role in ER stress may be specific for hepatocytes.

### 2.3. HPS Inhibits ER Stress-Induced ER Calcium Loss in Hepatocytes

TG is a calcium ion pump inhibitor that depletes calcium storage from the ER lumen by binding to sarcoplasmic/endoplasmic reticulum calcium ATPase (SERCA) on the ER membrane to form a conformation incapable of binding to calcium ions [24]. Calcium depletion in the ER lumen inactivates chaperones involved in protein folding and thereby triggers ER stress [26]. In addition, when calcium ions are released from the ER into the cytosol, it activates CaMKII in the cytosol, leading to CHOP-induced apoptosis [27]. Therefore, the modulation of calcium influx release from ER is an important way to maintain the balance of ER stress [28]. We explored the effects of HPS on calcium influx release from ER in primary hepatocytes using the fluorescent calcium probe Fluo-8 AM. The hepatocytes were incubated in a calcium-free medium, and ER calcium store depletion was induced by the addition of TG. The initial rise in cytosolic calcium induced by TG, which reflects the ER calcium content, was significantly lower in HPS-KO hepatocytes compared with WT hepatocytes (Figure 3A). After 8 h of TM treatment, ER calcium stores were markedly reduced in HPS-KO hepatocytes compared to WT hepatocytes (Figure 3B). We also performed a loss-of-function study using HPS-siRNA in mouse hepatocyte line AML12 cells. Compared with scrambled-siRNA, the siHPS knockdown of HPS significantly reduced TG-induced calcium release, reflected by the decreased ER calcium store (Figure 3C). The knockdown of HPS greatly decreased TG-sensitive calcium release in TM-treated AML12 cells, and this decrease was markedly reversed by HPS administration (Figure 3D). We examined the mRNA levels of genes related to Ca^2+^ influx, including *Serca*, *Stromal interaction molecule 1* (*Stim1*), *Na/Ca exchanger* (*Ncx*), *Inositol 1,4,5-trisphosphate receptor* (*Ip3r*) and *ORAI calcium release-activated calcium modulator 1* (*Orai*) [29]. An obvious decreased expression of *Serca* and *Orai* manifested in HPS-KO hepatocytes compared with WT (Figure 3E). These data indicate that HPS might be required for normal ER calcium handling in hepatocytes under steady-state conditions and be involved in regulating calcium influx release from ER in hepatocytes upon challenge with ER inducer.

### 2.4. HPS Deficiency Exacerbates Fasting/Refeeding-Induced Hepatic ER Stress in Mice

We evaluated the regulatory effect of HPS on physiological hepatic ER stress using the fasting/refeeding mice model [30], and found that plasma HPS levels increased with 16 h fasting and further increased after refeeding for 2 h (Figure 4A). The increased HPS in plasma after refeeding is unlikely to be due to fasting induction, as the half-life of circulating HPS in mice is 1.5 h [31]. Similarly, hepatic HPS protein was increased after 16 h of fasting and up to 4 h after refeeding (Figure 4B). Consistent with previous reports [30], refeeding after fasting induced hepatic ER stress gene expression in both WT and HPS-KO mice, which was further enhanced by HPS deficiency (Figure 4C). Moreover, the protein expressions of GRP78 and eIF2α phosphorylation in HPS-KO mice liver were induced by refeeding were higher than those of WT mice (Figure 4D). These results suggest that HPS may be involved in the regulation of physiological ER stress in the liver.

### 2.5. Deletion of HPS Sensitizes Mice to Severe Hepatic ER Stress Induced by TM

The influence of HPS deficiency on hepatic ER stress was determined in vivo using an intraperitoneal injection of TM in mice, which is a well-established ER stress-inducing agent that functions by inhibiting protein N-linked glycosylation [32]. We thus employed this model to determine the responses to ER stress of HPS-KO mice and their WT littermates at 24 h, 48 h and 72 h after 1 mg/kg TM treatment. The basic transcriptional activation of the UPR genes was compared between HPS-KO liver and WT liver, indicating normal UPR in HPS-KO mouse liver (Figure 5A). Twenty-four hours after TM injection, HPS-KO mice and control mice demonstrated robust transcriptional activation of the UPR markers in liver, but the degree of this was much higher in HPS-KO liver (Figure 5A). At 72 h following TM treatment, the expression of most hepatic ER stress genes in WT mice returned to the baseline levels, indicating a quick relief from ER stress. In turn, HPS-KO mice still displayed an elevation in these genes, consistent with ongoing UPR activation (Figure 5A). Immunoblotting analyses showed marked increases in GRP78 and CHOP protein levels, and the phosphorylation of eIF2α in HPS-KO liver at 24 h and 48 h after TM treatment (Figure 5B,C), confirming that the deletion of HPS makes mice sensitive to severe hepatic ER stress induced by TM.

### 2.6. HPS-KO Mice Show Aggravated TM-Induced Liver Injury

As a model of acute ER stress, the injection of TM induced the UPR to upregulate lipid synthesis genes as well as to downregulate the expression of β-oxidative genes, leading to hepatic steatosis [33]. WT mice showed mild hepatic steatosis at 24 h after a single dose of TM, as indicated by increased hepatic triglyceride contents, decreased serum triglyceride levels (Figure 6A), enhancement of Oil-Red-O (ORO) staining and its quantification (Figure 6B), whereas HPS-KO mice showed more aggravated hepatic steatosis when exposed to TM (Figure 6A,B). Histological analyses showed that there was markedly increased hepatocyte swelling and inflammatory infiltrate in HPS-KO mice liver (Figure 6B). Immunoblotting analyses showed that Bax expression and activated caspase 9 were increased in the livers of HPS-KO mice with TM, but Bcl-XL expression was decreased (Figure 6C), indicating that HPS deficiency may promote TM-induced apoptosis in liver. The mRNA expression of *interleukin-6* (*Il-6*) and *tumor necrosis factor-alpha* (*Tnfα*) in HPS-KO mice liver was higher than that of WT mice (Figure 6D). Moreover, the serum alanine aminotransferase (ALT) activity was greater in HPS-KO mice challenged with TM (Figure 6E). These results suggest that HPS deletion may aggravate the TM-induced inflammatory response and worsens liver injury. In addition to its hepatotoxicity, TM is also able to induce acute kidney injuries in intoxicated animals [34]. We found that TM-induced increases in blood urea nitrogen (BUN) and the mRNA abundance of ER stress marker genes in the kidneys were comparable in both genotypes of mice (Appendix A), indicating that HPS inactivation might selectively affect TM-induced ER stress in the liver.

To explore the mechanisms by which HPS deletion exacerbates TM-induced hepatic steatosis, we examined the mRNA levels of genes related to fatty acid metabolism in the liver, involving fatty acid absorption, de novo synthesis, oxidation, and transport. The similar expressions of *CD36*, *fatty acid transporter protein 2, 5* (*Fatp 2*, *5*) and lipid export proteins, such as *microsomal triglyceride transfer protein* (*Mttp*) between the livers of HPS-KO mice and their WT controls following TM injection were observed, whereas the increased expression of *sterol regulatory element-binding protein 1* (*Srebp-1c*) and its target *fatty acid synthase* (*Fasn*), *acetyl-CoA carboxylase alpha* (*Acc1*) and *stearoyl-coenzyme A desaturase 1* (*Scd1*), and decreased expression of *carnitine palmitoyl-CoA transferase 1α* (*Cpt1α*) and *peroxisome proliferator-activated receptor alpha* (*PPARα*) manifested in TM-treated HPS-KO liver compared with control liver (Figure 6F). These results suggest that HPS may not influence fatty acid absorption and transport, but affect lipid synthesis and β-oxidation under ER stress.

### 2.7. HPS Administration Reverses Aggravated Hepatic Steatosis in HPS-KO Mice during TM-Induced ER Stress

We further performed rescue assays in TM-treated HPS-KO mice with HPS protein administration. The preliminary administration of HPS to HPS-KO mice inhibits the TM-induced hepatic mRNA expression levels of genes for ER stress to the levels seen in TM-treated WT mice (Figure 7A). Immunoblotting analyses confirmed that the administration of HPS significantly blocks the TM-induced activation of ER stress signaling in HPS-KO mice liver, as evidenced by the decreased expressions of GRP78 and CHOP, the phosphorylation of eIF2α, and the nuclear accumulation of SREBP1c (Figure 7B). Specifically, TM-induced lipid accumulation in hepatocytes was significantly diminished in HPS-KO mice through the application of HPS, as determined by staining with ORO and its quantification and measuring triglyceride contents (Figure 7C,D). Serum ALT levels were much lower in HPS-treated HPS-KO mice than control mice, which were similar to TM-treated WT mice, consistent with alleviated liver injury (Figure 7E). Furthermore, TM-induced liver injury in WT mice was also improved by HPS pre-treatment (Figure 7C–E). To evaluate the therapeutic potential of HPS, we intraperitoneally injected HPS into mice after TM injection. Six hours after injection of TM in WT and HPS-KO mice, hepatic ER stress genes were significantly up-regulated and serum triglyceride and TCHO were significantly decreased in both genotypes, but only HPS-KO mice had elevated serum ALT levels (Appendix A). The injection of HPS can significantly reduce TM-induced serum ALT levels and hepatic lipid accumulation in WT and KO mice, which were similar to ER inhibitors 4-phenylbutyric acid (PBA) and tauroursodeoxycholic acid (TUDCA) (Figure 8A,B) [35]. However, unlike PBA and TUDCA, the injection of HPS did not reduce serum BUN and creatine kinase isoenzyme (CK) levels (Figure 8A), which is a marker for cardiac function evaluation as severe ER stress can result in heart failure [36]. Our data suggest a potential protective role of HPS in ER stress-induced acute liver injury.

### 2.8. Exacerbated Hepatic ER Stress Contributes to Severe Steatohepatitis Induced by MCD Diet in HPS-KO Mice

Feeding mice an MCD diet could suppress the production of very low density lipoproteins by inhibiting the synthesis of phospholipids in the liver, which leads to the deposition of triglycerides in the liver, resulting in hepatic steatosis [16]. It has been reported that a positive feedback loop exists between lipid accumulation and ER stress [33]. Our previous studies showed that HPS-KO mice exhibited more pronounced ER stress in the MCD feeding model, which is considered a chronic ER stress model [16]. We gave mice an intraperitoneal injection of the ER stress inhibitor PBA or TUDCA after feeding the mice with MCD diet for 3 weeks [37,38]. Steatohepatitis was significantly improved in HPS-KO mice as assessed by serum ALT (Figure 9A) and hepatic triglyceride measurement (Figure 9B), and histology analysis (Figure 9C). Consistent with the previous observation [37], PBA treatment only slightly improved MCD-induced steatohepatitis in WT mice (Figure 9A–C). TUDCA also showed a similar inhibitory effect (Figure 9D–F). These results indicate that *HPS* deletion may accelerate MCD diet-induced steatohepatitis via activating ER stress.

## 3. Discussion

In this paper, we demonstrated a negative feedback mechanism for the regulation of ER stress by HPS in hepatocytes. In this mechanism, HPS expression is upregulated after ER stress through the PERK-ATF4-CEBPB cascade; in turn, the induced HPS acts to restrict the ER stress response at least partly by modulating cellular calcium homeostasis, and contributes to the maintenance of the balance of intrahepatic ER stress under physiological and pathological conditions. Because adult HPS-deficient mice liver is histologically normal under steady-state conditions [13,16], HPS may not be necessary to maintain the liver ER stress balance under physiological conditions, but it contributes to stress conditions. Several other genes involved in hepatic ER stress regulation play a similar role, and their knockout mice show little or no liver phenotype under normal conditions [39,40].

HPS has emerged as an acute phase reactant, as its expression in the liver is induced by different kinds of stresses [12,41,42]. Here, we confirmed that HPS may function as an ER stress response protein. During ER stress, the PERK, IRE1 and ATF6 signaling pathways are activated to initiate the transcription of many genes [43]. Through pharmacological inhibition, we determined that the PERK cascade regulated *HPS* transcription in hepatocytes’ ER stress response. Under ER stress, PERK activates eIF2α through autophosphorylation and oligomerization to increase ATF4 expression [44]. ATF4 plays an important role in the regulation of various physiological events by activating a great number of target genes downstream from the PERK pathway [45]. We found that the overexpression of ATF4 mimicked ER stress-induced *HPS* promoter activity, but failed to induce the *HPS* promoter lacking the binding site of CEBPB, which is a target gene of ATF4 and upregulated following TM treatment. A previous report showed that CEBPB bound directly to HPS promoter and activated HPS transcription [21]. We thus speculate that ER stress increases the expression of CEBPB by activating the PERK/ATF4 pathway, and then CEBPB may upregulate *HPS* expression at the transcriptional level by directly binding to its promoter.

ER stress pathways have been critically linked to the regulation of the expression of lipogenic factors and facilitate the development of hepatosteatosis [46,47,48]. In line with these studies, TM treatment is reported to increase lipogenesis, inhibit fatty acid oxidation and reduce triglyceride output, leading to hepatosteatosis [30]. Here, we demonstrated that HPS deficiency exacerbated TM-induced hepatic steatosis largely through the enhancement of de novo synthesis and the impairment of the β-oxidation of fatty acids. In addition to regulating lipid metabolism, cell death and inflammation are two other outputs of ER stress. The excessive accumulation of unfolded proteins in ER leads to the oligomerization and autophosphorylation of IRE1, which activates the Apoptotic-Signaling Kinase-1 (ASK1)-c-Jun NH2-terminal kinase (JNK)-Bim or interacts with PERK-ATF4-CHOP pathways, ultimately promoting apoptosis [44,49]. Here, we demonstrated that TM treatment activated CHOP and PERK pathways more strongly in HPS-KO liver. Meanwhile, HPS deficiency increased proapoptotic factor Bax expression and decreased antiapoptotic factor Bcl-XL expression in HPS-KO mice liver with TM. These results suggest that loss of HPS may promote hepatocyte apoptosis in TM-treated mice. TM treatment also increased the expressions of *Il-6* and *Tnfα* in HPS-KO mice liver, indicating more inflammatory responses in the liver tissues of HPS-KO mice. These data suggest that disturbed lipid metabolism and increased cell death and inflammation may exacerbate liver injury in HPS-KO mice treated with TM.

Although more studies are required to carefully assess the precise mechanism of HPS in the regulation of ER stress, the evaluation of ER calcium release into the cytosol suggested that HPS deficiency might decrease the basal ER calcium levels in hepatocytes. This change might not be due to the stronger ER stress response in HPS-KO hepatocytes under steady-state conditions. One possible explanation for this is that HPS may be involved in the regulation of calcium entry or intracellular calcium handling by the ER. It is well known that a decrease in ER calcium activates the ER stress response, as the optimal activity of many chaperones in ER depends on calcium [25,50,51]. On the other hand, we found that TM-induced ER calcium depletion was blocked by HPS, implying that HPS might attenuate ER stress pathways through the suppression leakage of ER calcium. HPS is secreted extracellularly by hepatocytes via specific signal peptides, stimulates receptors on the membrane, and activates the ERK1/2 pathway to promote hepatocyte proliferation [11]. Interestingly, ERK1/2 can regulate the expression of the calcium pump SERCA by phosphorylating CREBP [52]. In addition, ERK1/2 is also involved in the phosphorylation of STIM1 and its interaction with Orai, thereby influencing calcium influx [53]. We here identified the downregulation of Orai in HPS-KO hepatocytes. We therefore speculate that an imbalance in ER calcium homeostasis occurs in HPS-KO hepatocytes may be due to the dysregulation of ERK1/2 -mediated ER calcium regulatory signal. Further study on the relationship between the ERK1/2 activation induced by HPS and ER calcium homeostasis will help to reveal the molecular mechanism of HPS in regulating ER stress.

Excessive and sustained ER stress is an important contributor to the progression of many diseases, such as diabetes, tumors, NAFLD, and aging [1,2,44,54]. Over nutrition or aging can induce insulin resistance, which will lead to the lipolysis of white adipose tissue, thus increasing saturated fatty acids in blood. Excess fatty acids are transported into hepatocytes and directly inserted into the ER, resulting in reduced membrane fluidity and the activation of UPR [55]. Unresolved UPR leads to lipid accumulation and apoptosis, promoting steatosis and liver injury [46,47,48,49]. In our study, the administration of HPS significantly alleviated TM-induced liver injury even when it was administered up to 6 h after TM treatment, which suggest that HPS may be used for treatment rather than just for the prevention of ER stress-induced liver injury. We previously demonstrated that HPS deficiency led to elevated hepatic ER stress signaling activity in mice on an MCD diet [16], but its exact role has not been determined. Here, we found that the ER stress inhibitor improved MCD diet-induced steatohepatitis in HPS-KO mice, indicating a potential hepatoprotective role of endogenous HPS in chronic ER stress. Considering that HPS has an important protective role in glucotoxicity- and lipid toxicity-induced liver injury [16,19,20], our findings suggest that the maintenance of intrahepatic ER stress balance may be a general hepatoprotective mechanism of HPS. Drugs that improve ER stress by promoting adaptive UPR signal transduction offer therapeutic opportunities for liver injury [54]. Thus, HPS may be effective for the management of ER stress-related liver diseases, such as NAFLD. It would be of much interest to study the effect of HPS in protecting the liver in the progression of various human liver diseases by resisting ER stress. Considering the metabolic properties of recombinant HPS proteins with a shorter metabolic time in vivo [30], it would be more helpful to develop small-molecule analogues that mimic the structure of HPS or to target HPS for the development of agonists.

In summary, the present study reveals a previously unknown role of HPS in protecting ER stress-induced liver injury, and proposes that HPS may be a novel potential treatment for ER stress-related liver disease.

## 4. Materials and Methods

### 4.1. Animal Experiments

Eight-week-old, male, HPS-deficient (HPS-KO) mice and their WT littermates have been described previously [16]. ER stress was induced by intraperitoneal injection with tunicamycin (TM, Sigma-Aldrich, St. Louis, MO, USA) dissolved in buffer (1% DMSO/150 mM glucose) at 1.0 mg/kg of body weight. HPS-KO or WT mice with TM were administrated intraperitoneally with HPS (1 mg/kg) or 4-phenylbutyric acid (PBA, 100 mg/kg) (Sigma-Aldrich, St. Louis, MO, USA) or tauroursodeoxycholic acid (TUDCA, 250 mg/kg) (Calbiochem, La Jolla, CA, USA) or phosphate-buffered saline (PBS) at 1 h before or 6 h after TM injection. After being fed methionine- and choline-deficient (MCD) (Research Diet, A02082002BR, New Brunswick, NJ, USA) diet for three weeks, WT or HPS-KO mice were injected intraperitoneally with PBA (100 mg/kg) every three days for one week or TUDCA (250 mg/kg) every day for one week [16]. Serum triglyceride, total cholesterol (TCHO), urea nitrogen (BUN), creatine kinase isoenzyme (CK), alanine transaminase (ALT) and liver triglyceride were measured as previously described [16]. Histological analysis was performed using standard procedures, as described previously [16]. Haematoxylin and eosin (HE) histological liver slides were scored using the previously described liver steatosis grade [56]. The Oil-Red-O (ORO) staining area fraction was quantitated using ImageJ software and the values are the mean ± SD of 3–5 slides of measurements. All mice were housed in a specific pathogen-free (SPF) environment with free access to food and water. The animal facility was under a 12 h light–dark cycle. All animal experiments were designed according to the guidelines set by the Institutional Animal Care and Use Committee of our institution (IACUC-2020-567, IACUC-DWZX-2022-611).

### 4.2. Primary Hepatocytes and Cell Lines

A two-step collagenase perfusion technique was used to isolate primary murine hepatocytes from WT and HPS-KO mice [57]. Human hepatoma cell line HepG2 and mouse hepatic cell line AML12 were obtained from the Cell Center of the Chinese Academy of Medical Sciences (Beijing, China), and maintained in Dulbecco’s modified Eagle’s medium (DMEM) or RPMI Medium 1640 (Gibco, Carlsbad, CA, USA) supplemented with 10% fetal bovine serum (MD, St. Louis, MO, USA) according to the manufacturer. SiRNAs targeting HPS (sc-62453, Santa Cruz Biotechnology (Shanghai) Co., Ltd., Shanghai, China) were transfected into AML12 cells to knockdown HPS protein expression.

For pharmacological treatment, cells were given 100 or 400 ng/mL HPS protein, 15 µM thapsigargin (TG; Sigma-Aldrich, St. Louis, MO, USA) or 4 μg/mL TM.

### 4.3. Measurement of ER Calcium Levels in Hepatocytes

Calcium levels in ER were assessed by measuring the release of calcium from the ER after TG treatment as described previously [22]. Briefly, 5 × 10^4^ cells were seeded in a 24-well plate, and after adhering and extending, the cells were exposed to 4 μg/mL TM for 8 h or left untreated. Then, the cells were washed with HBSS buffer without calcium and magnesium ions three times, and incubated with Fluo-8 AM (Abcam, San Diego, CA, USA) for 30 min. After washing cells three times with calcium-free HBSS, steady-state cytosolic calcium levels were measured for 1 min at 23 °C. ER calcium store depletion was induced by 2 μM TG treatment and the calcium fluorescence was detected at Ex/Em 490/525 nm. Calcium release values were normalized to the background levels.

### 4.4. Measurement of Luciferase Activities In Vitro and In Vivo

In vitro and in vivo transfections were performed as previously described [58]. TM was used for 24 h to induce ER stress 16 h after transfection. The luciferase activities of the *HPS* promoter construct (HPS-luc) were detected using the dual-Luciferase^TM^ reporter assay system (Promega, Madison, WI, USA). The results are presented as the fold induction relative to the values from the control group transfected with pGL3-basic.

### 4.5. Measurement of Cytotoxicity

A total of 4 × 10^5^ cells were seeded in 6-well plates and subjected to pharmacological treatment. Cell culture supernatants were collected for the detection of LDH release levels by colorimetry using the LDH Cytotoxicity Assay Kit (E1020, Applygen Technologies Inc., Beijing, China). The percentages of apoptosis were analyzed by flow cytometry using a fluorescein isothiocyanate (FITC) Annexin V Apoptosis Detection Kit with propidium iodide (PI) (640914, Biolegend, San Diego, CA, USA).

### 4.6. Measurement of Cell Viability

The cell survival rate was evaluated using an MTT assay (Abcam, Cambridge, UK, ab211091) according to the manufacturer’s instructions.

### 4.7. Immunoblot Analysis

The nucleus and cytoplasm protein were isolated from approximately 50 mg of liver tissue or 1 × 10^6^ cells and subjected to immunoblot analysis after quantification, as previously described [16,58]. The antibodies are listed in Appendix A.

### 4.8. Real-Time Quantitative PCR Analysis

Total RNA was extracted from liver tissues or cells using Trizol reagent and 5 μg RNA was used for cDNA synthesis according to the manufacturer’s protocol. Real-time quantitative PCR was carried out with specific primers for each gene (Appendix A) [16,23,59]. Relative mRNA levels were normalized to *TATA-box binding protein* (*TBP*), which was used as a housekeeping gene.

### 4.9. Statistical Analysis

Data from in vivo experiments are expressed as mean ± SD. Data from in vitro experiments are expressed as mean ± SEM. GraphPad Prism (version 8; GraphPad Software, San Diego, CA, USA) was used to draw graphs and statistics. Student’s *t*-test was used to compare two data sets. Two-way ANOVA was used for multiple group comparisons. *p* < 0.05 was considered statistically significant.

## Figures and Tables

**Figure 1 ijms-23-13325-f001:**
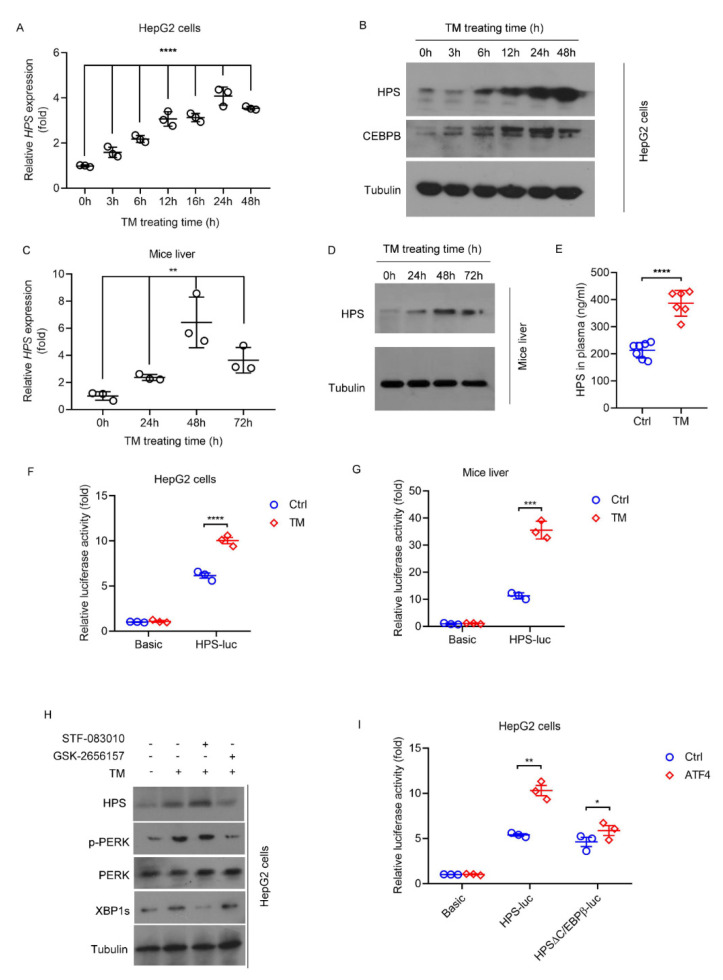
ER Stress induces HPS expression in hepatocytes through PERK-ATF4 pathway. For panels (**A**,**B**), HepG2 cells were treated with TM (4 μg/mL) for the indicated number of hours. (**A**) Relative expression of *HPS* mRNA was measured by q-PCR. Two-way ANOVA was used for statistics (F = 59.44, **** *p* < 0.0001) (**B**) Immunoblotting of protein levels of HPS and CEBPB. For panels (**C**–**E**), 8-week-old WT C57BL/6J male mice were treated with TM (1 mg/kg body weight) intraperitoneal injection for the indicated number of hours. (**C**) Relative expression of *HPS* mRNA was measured by q-PCR (*n* = 3). Two-way ANOVA was used for statistics (F = 17.82, ** *p* = 0.0022) (**D**) The liver protein samples of 3 mice in each group were pooled for immunoblotting analysis to detect the levels of HPS. (**E**) ELISA was performed to measure HPS levels in plasma of mice (*n* = 6–7). (**F**) HepG2 cells were transfected with *HPS* promoter construct (HPS-luc) and treated with TM (4 μg/mL) for 24 h. (**G**) *HPS* promoter construct (HPS-luc) was transfected into the mouse liver by hydrodynamic method and treated with TM (1 mg/kg body weight) intraperitoneal injection for 24 h (*n* = 3). (**H**) Immunoblotting of protein levels of HPS, p-PERK, PERK and XBP1s in HepG2 cells after TM and ER stress inhibitor treatments. Cells were treated with STF-083010 (IRE1α inhibitor, 50 μM) or GSK-2656157 (PERK inhibitor, 1 μM) for 1 h, followed by TM (4 μg/mL) for 24 h. (**I**) HepG2 cells were transfected with pGL3/basic or *HPS* promoter construct (HPS-luc) or HPSΔCEBPB-luc in combination with control vector or ATF4-overexpressing construct. For panel (**F**,**G**,**I**), dual-luciferase reporter assays were used to analyze *HPS* promoter activities. Data from in vivo experiments are expressed as mean ± SD of all samples for each group. Data from in vitro experiments are expressed as mean ± SEM of three independent experiments (two duplicates per experiment). For panels (**E**–**G**), Student’s *t* test was used to compare the mean relative values between TM group and control groups (*** *p* < 0.001, **** *p* < 0.0001). For panel (**I**), Student *t* test was used to compare the mean relative values between ATF4 overexpression group and ctrl group (* *p* < 0.05, ** *p* < 0.01).

**Figure 2 ijms-23-13325-f002:**
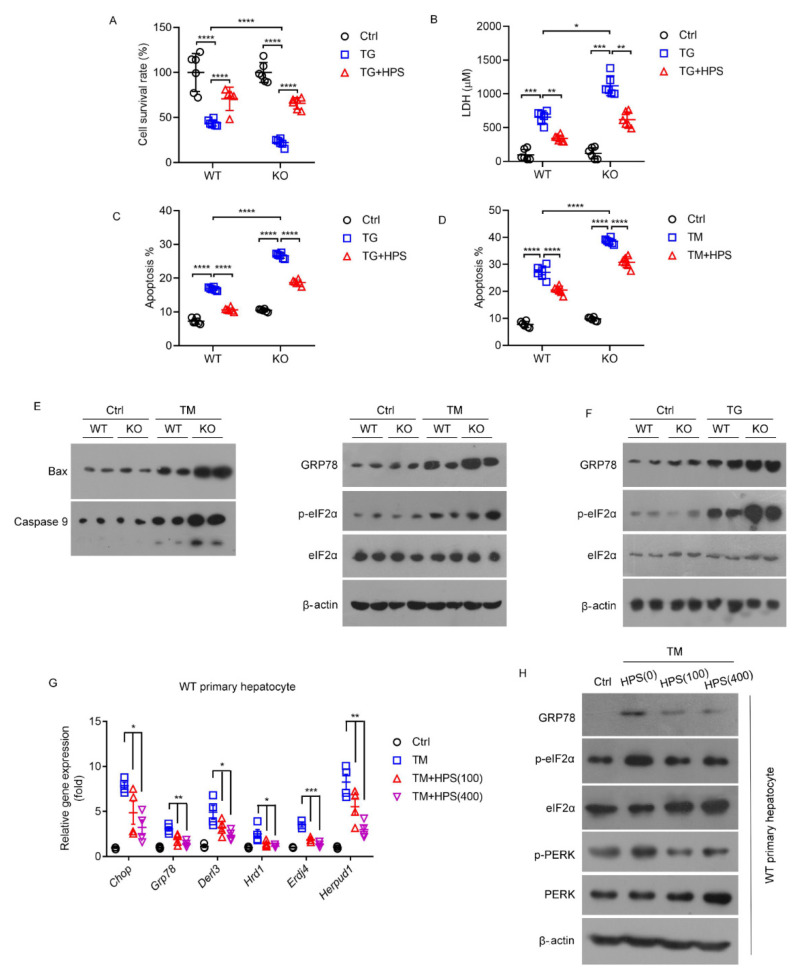
HPS is responsible for ER stress protection in hepatocytes in vitro. For panel (**A**–**C**), WT or HPS-KO primary hepatocytes were treated with HPS (400 ng/mL, 8 h) and TG (15 μM, 5 h). (**A**) MTT was performed to detect cell survival. (**B**) ELISA was performed to determine LDH levels in primary hepatocytes culture media. (**C**) Flow cytometry was performed to analyze percentages of apoptosis. For panels (**D**–**F**) WT or HPS-KO primary hepatocytes were treated with HPS (400 ng/mL, 2 h) and TM (4 μg/mL, 12 h). (**D**) Flow cytometry was performed to analyze percentages of apoptosis. (**E**,**F**) Immunoblotting of the protein levels of pro-apoptotic protein (BAX and CASPASE 9) and ER stress marker (GRP78, p-eIF2α and eIF2α). Q-PCR (**G**) and immunoblot (**H**) analysis of ER stress markers in WT primary hepatocytes treated with HPS (100 or 400 ng/mL, 2 h) and TM (4 μg/mL, 12 h). Data are expressed as mean ± SEM of at least three independent experiments (two duplicates per experiment). For panel (**A**–**D**), the significant difference between the mean relative values of two groups was analyzed by Student *t* test (* *p* < 0.05, ** *p* < 0.01, *** *p* < 0.001, **** *p* < 0.0001). For panel (**G**), two-way ANOVA was used for statistics (*Chop*: F = 59.44, * *p* < 0.0126; *Grp78*: F = 47.16, *** *p* = 0.0002; *Derl3*: F = 18.28, ** *p* = 0.0028; *Hrd1*: F = 10.36, * *p* = 0.0113; *Erdj4*: F = 23.49, ** *p* = 0.0062; *Herpud1*: F = 57.52, *** *p* = 0.0001).

**Figure 3 ijms-23-13325-f003:**
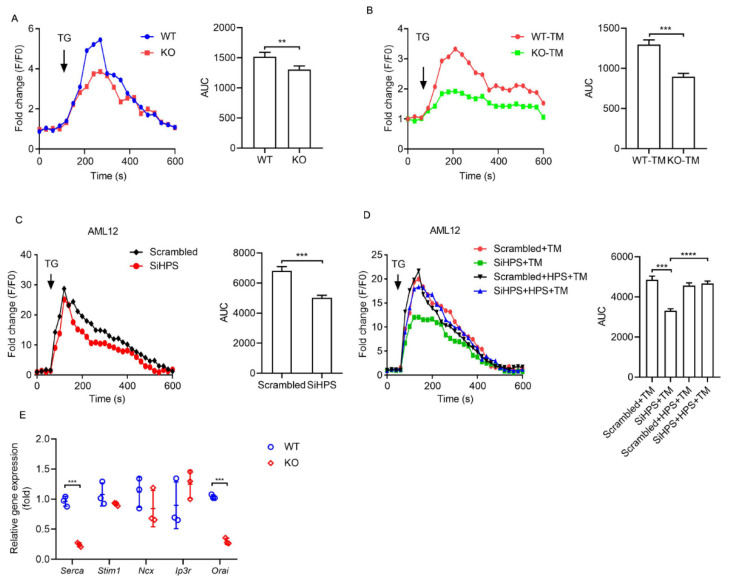
HPS inhibits ER stress-induced ER calcium loss in hepatocytes. (**A**) ER calcium store depletion by 2 μM TG treatment in WT or HPS-KO primary hepatocytes. (**B**) WT or HPS-KO primary hepatocytes were exposed to 4 μg/mL TM for 8 h followed by inducing calcium release with 2 mM TG treatment. For panels (**C**,**D**), AML12 cells were transfected with HPS siRNA (SiHPS) or nonspecific RNA duplex (Scrambled) for 72 h. (**C**) Calcium release was induced by 2 µM TG. (**D**) The cells were incubated with 250 ng/mL HPS for 2 h followed by treating with 4 μg/mL TM for 8 h. Calcium release was induced by 2 µM TG. The calcium fluorescence was detected at Ex/Em 490/525 nm. The results are presented as fold change relative to the F0. The AUC was used to determine the relative calcium levels. (**E**) Q-PCR analysis of the mRNA levels of genes related to Ca^2+^ influx in WT or HPS-KO primary hepatocytes. Data are expressed as mean ± SEM of three independent experiments (two duplicates per experiment). The significant difference between the mean relative values of two groups was analyzed by Student *t* test (** *p* < 0.01, *** *p* < 0.001, **** *p* < 0.0001).

**Figure 4 ijms-23-13325-f004:**
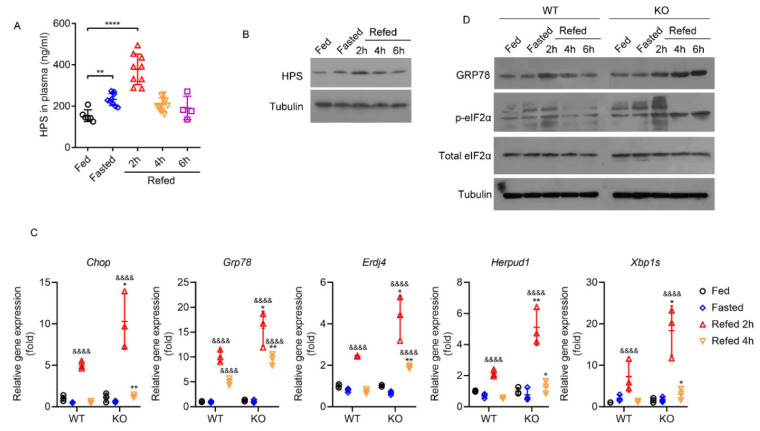
HPS deficiency exacerbates fasting/refeeding induced Hepatic ER stress in mice. For panels (**A**,**B**), eight-week-old WT mice were starved for 16 h, followed by refeeding for the number of hours indicated (*n* = 3–12). (**A**) ELISA was performed to measure plasma HPS levels. Data are expressed as mean ± SD of all samples for each group. The significant difference between the mean relative values of the two groups was analyzed by Student *t* test (** *p* < 0.01, **** *p* < 0.0001). (**B**) The liver protein samples of three mice in each group were pooled for immunoblotting analysis to detect the levels of HPS. For panels (**C**,**D**), 8-week-old WT and HPS-KO mice were starved for 16 h, followed by refeeding for the number of hours indicated (*n* = 3). (**C**) Q-PCR analysis of ER stress markers in mice liver. Data are expressed as mean ± SD of all samples for each group. The significant difference between the mean relative values of two groups was analyzed by Student *t* test (&&&& *p* < 0.0001 versus corresponding Fed group; * *p* < 0.05, ** *p* < 0.01 versus corresponding WT group) (**D**) The liver protein samples of three mice in each group were pooled for immunoblotting analysis to detect the levels of ER stress markers.

**Figure 5 ijms-23-13325-f005:**
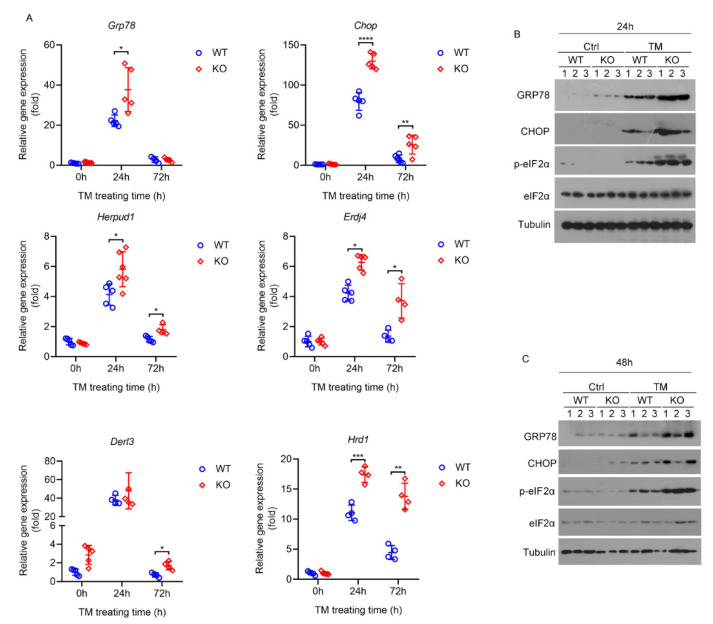
Deletion of HPS sensitizes mice to severe hepatic ER stress induced by TM. Eight-week-old WT and HPS-KO mice were injected intraperitoneally with TM (1 mg/kg body weight). (**A**) Relative expression of ER stress gene (*Grp78*, *Chop*, *Derl3*, *Hrd1*, *Herpud1* and *Erdj4*) mRNA in liver of the mice treated with TM for 24 h or 72 h was measured by q-PCR (*n* = 4–6). (**B**,**C**) Immunoblotting analysis of GRP78, CHOP, p-EIF2α and EIF2α levels in liver of the mice treated with TM for the number of hours indicated (*n* = 3). Arabic numeral represents the number of the mouse to which the protein sample belongs in each group. Data are expressed as mean ± SD of all samples for each group. The significant difference between the mean relative values of two groups was analyzed by Student *t* test (* *p* < 0.05, ** *p* < 0.01, *** *p* < 0.001, **** *p* < 0.0001).

**Figure 6 ijms-23-13325-f006:**
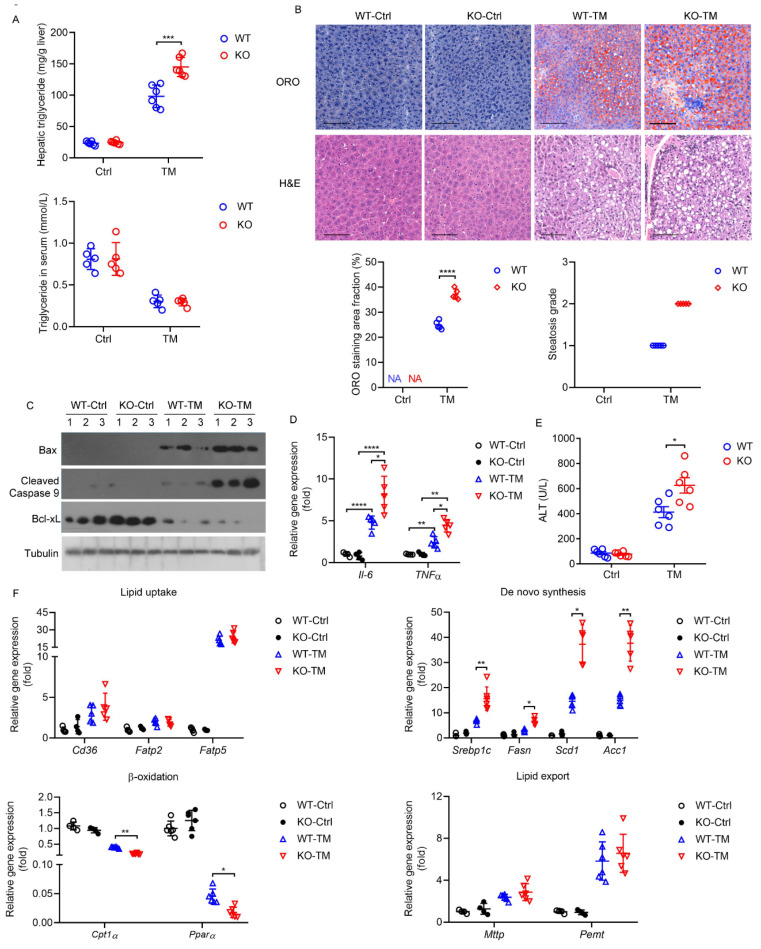
HPS-KO mice show aggravated TM-induced hepatic steatosis. Eight-week-old WT and HPS-KO mice were injected intraperitoneally with TM (1 mg/kg body weight) for 24 h. (**A**) Hepatic and serum triglyceride levels of the mice (*n* = 5–6). (**B**) Representative images of ORO- and H and E-stained liver sections of the mice (*n* = 6). Scale bar = 100 μm. The ORO staining area fraction was quantitated using ImageJ software and values are the mean ± SD of 5 liver sections of measurements. The steatosis grade was evaluated according to steatosis area fraction. <5%, score 0; 5–33%, score 1; 33–66%, score 2; >66%, score 3. (**C**) Immunoblotting analysis of protein levels of Bax, cleaved Caspase 9 and Bcl-xL in mice liver. Arabic numeral represents the number of the mouse to which the protein sample belongs in each group. (**D**) Relative expression of *Il6* and *Tnfα* mRNA in mice liver was measured (*n* = 4–6). (**E**) Serum ALT levels of the mice (*n* = 6). (**F**) Relative expression of *Cd36*, *Fabp2*, *Fabp5*, *Srebp1c*, *Fasn*, *Scd1*, *Acc1*, *Mttp*, *Cpt1α* and *Pparα* mRNA in mice liver was measured (*n* = 4–6). Data are expressed as mean ± SD of all samples for each group. The significant difference between the mean relative values of two groups was analyzed by Student *t* test. (* *p* < 0.05, ** *p* < 0.01, *** *p* < 0.001, **** *p* < 0.0001).

**Figure 7 ijms-23-13325-f007:**
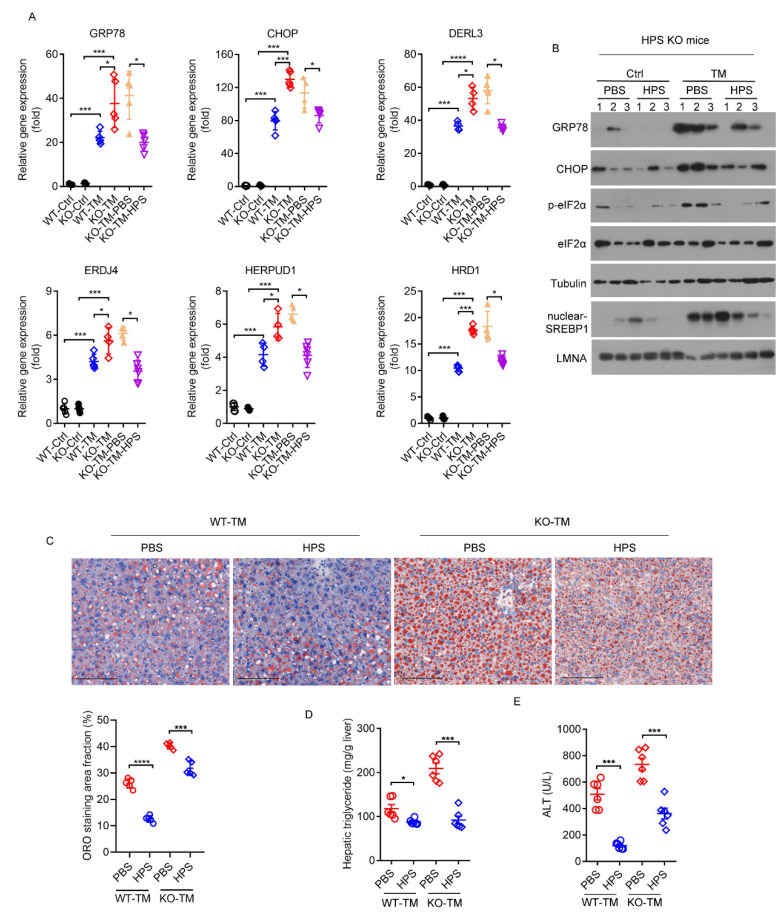
Pre-treatment HPS inhibits aggravated hepatic steatosis in HPS-KO mice during TM-induced ER stress. Eight-week-old WT and HPS-KO mice were injected intraperitoneally with TM (1 mg/kg body weight) for 24 h. For the rhHPS-treated group, 1 h before intraperitoneal injection TM, WT or HPS-KO male mice were injected intraperitoneally with rhHPS (1 mg/kg) or PBS. (**A**) Relative expression of *Grp78*, *Chop*, *Derl3*, *Hrd1*, *Herpud1* and *Erdj4* mRNA in mice liver (*n* = 4–6). (**B**) Immunoblotting analysis of GRP78, CHOP, p-eIF2α and eIF2α in whole-cell lysates and SREBP1 in nuclear extracts. Tubulin was used as the whole-cell lysate control and LMNA was used as the nuclear extract control. The Arabic numeral represents the number of the mouse to which the protein sample belongs in each group. (**C**) Representative images of ORO-stained liver sections of the mice as indicated. Scale bar = 100μm. The ORO staining area fraction was quantitated using ImageJ software and values are the mean ± SD of five liver sections of measurements. (**D**) Hepatic triglyceride levels of the mice (*n* = 6). (**E**) Serum ALT levels of the mice (*n* = 6). Data are expressed as mean ± SD of all samples for each group. The significant difference between the mean relative values of two groups was analyzed by Student *t* test (* *p* < 0.05, *** *p* < 0.001, **** *p* < 0.0001).

**Figure 8 ijms-23-13325-f008:**
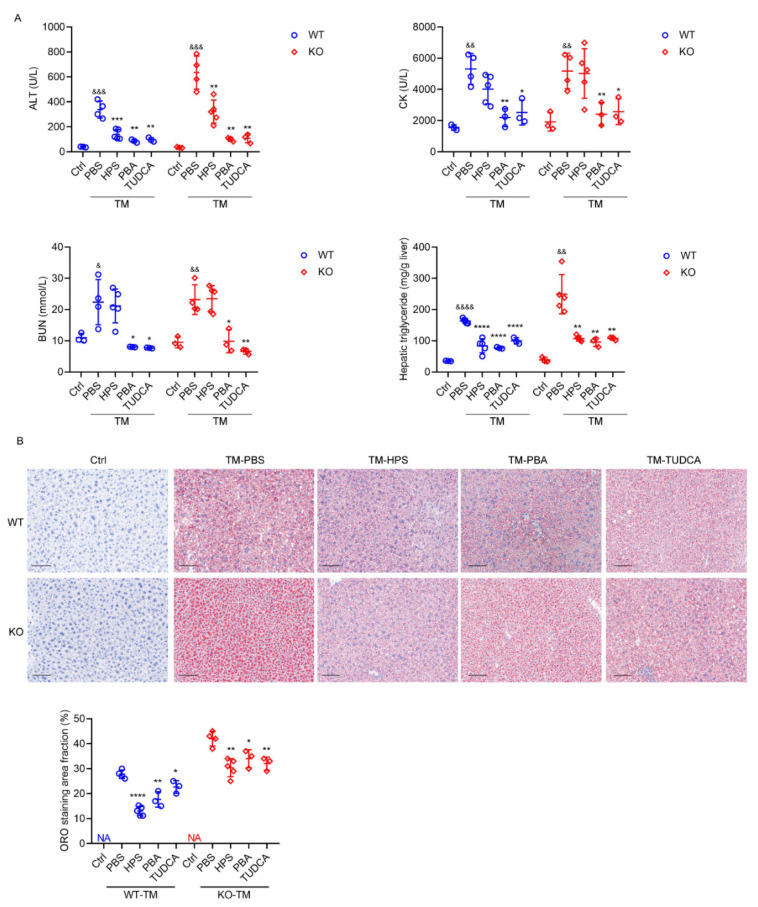
HPS ameliorate TM-induced hepatic steatosis. Eight-week-old WT and HPS-KO mice were injected intraperitoneally with TM (1 mg/kg body weight) for 24 h. Six hours after injection of TM, WT or HPS-KO male mice were injected intraperitoneally with rhHPS (1 mg/kg) or PBA (100 mg/kg) or TUDCA (250 mg/kg). (**A**) Serum ALT, BUN and CK levels and hepatic triglyceride levels of the mice (*n* = 3–5). (**B**) Representative images of ORO-stained liver sections of the mice as indicated. Scale bar = 100 μm. The ORO staining area fraction was quantitated using ImageJ software and values are the mean ± SD of 3–5 liver sections of measurements. Data are expressed as mean ± SD of all samples for each group. The significant difference between the mean relative values of two groups was analyzed by Student *t* test (* *p* < 0.05, ** *p* < 0.01, *** *p* < 0.001, **** *p* < 0.0001 versus corresponding TM-PBS; & *p* < 0.05, && *p* < 0.01, &&& *p* < 0.001, &&&& *p* < 0.0001 versus corresponding ctrl).

**Figure 9 ijms-23-13325-f009:**
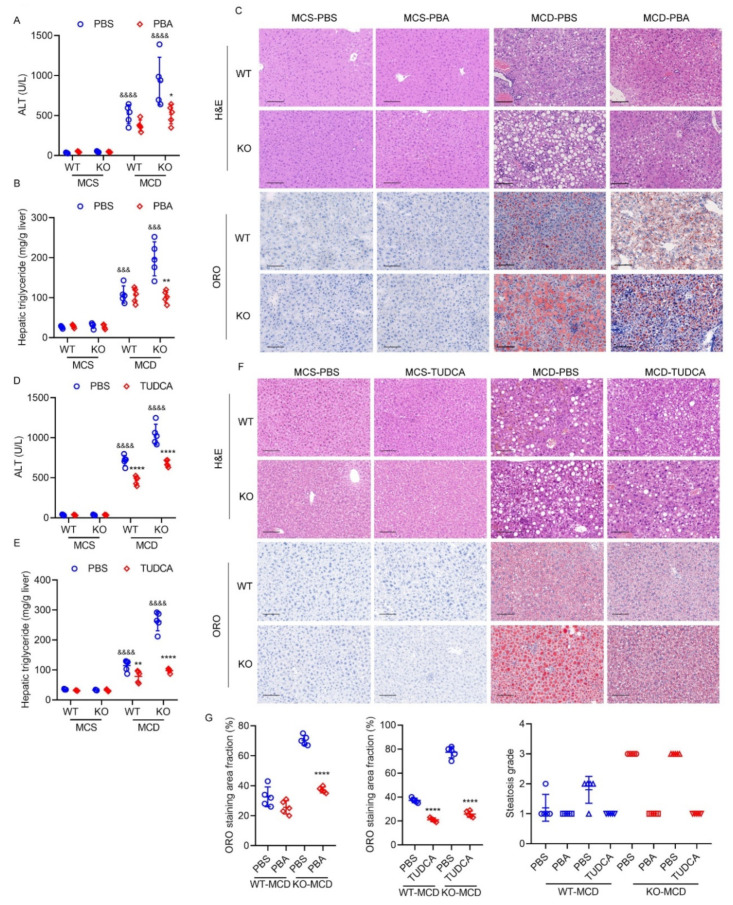
Exacerbated hepatic ER stress contributes to severe steatohepatitis induced by MCD diet in HPS-KO mice. Eight-week-old WT and HPS-KO mice were maintained on an MCD diet for three weeks. Then, the mice were divided into two groups with one group receiving intraperitoneal injection of PBS and another group receiving PBA (100 mg/kg body weight) every three days for one week or TUDCA (250 mg/kg body weight) every day for one week. During this period, the mice were still fed an MCD diet. (**A**,**D**) Serum ALT levels of the mice (*n* = 5). (**B**,**E**) Hepatic triglyceride content of the mice (*n* = 6). (**C**,**F**) Representative pictures of H and E and ORO staining (*n* = 5). Scale bar = 100 μm. (**G**) The ORO staining area fraction was quantitated using ImageJ software and values are the mean ± SD of 3–5 liver sections of measurements. The steatosis grade was evaluated according to the steatosis area fraction. <5%, score 0; 5–33%, score 1; 33–66%, score 2; >66%, score 3. Data are expressed as mean ± SD of all samples for each group. The significant difference between the mean relative values of two groups was analyzed by Student *t* test (* *p* < 0.05, ** *p* < 0.01, **** *p* < 0.0001 versus corresponding WT-MCD; &&& *p* < 0.001, &&&& *p* < 0.0001 versus corresponding ctrl).

## Data Availability

All data are contained within this paper.

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
