# Peer review of "Protective Role of Hepassocin against Hepatic Endoplasmic Reticulum Stress in Mice"

_ijms, 2022, doi:10.3390/ijms232113325_

Round 1
Reviewer 1 Report (Previous Reviewer 2)
Hepassocin (HPS) is a hepatokine that has multiple proposed physiological functions and some of the biological processes in which it is involved are closely related to endoplasmic reticulum (ER) stress, but the role of HPS in the regulation of ER stress remains unclear.
In this study, the authors aimed to demonstrate that Hepassocin (HPS) transcription was induced by the protein kinase RNA-like ER kinase (PERK)/activating transcription factor 4 (ATF4) cascade upon ER stress in hepatocytes. Overall, their results would support that HPS acts in a negative feedback fashion to regulate hepatic ER stress and protects hepatocytes from ER stress-related injury. HPS has the potential to be a candidate drug for the treatment of ER stress-related liver injury. The Authors concluded that results reveal a previously unappreciated role of HPS in protecting ER stress-induced liver injury, and proposes that HPS may be a novel potential treatment for ER stress-related liver disease.
The study is of interest and of potential clinical impact. However, in my opinion, the authors should stress the clinical implications of their findings. In particular, they should discuss evidences of ER-involvement in liver diseases.
Author Response
Please see the attachment.

Reviewer 2 Report (Previous Reviewer 1)
Authors have substantially improved the manuscript.
Author Response
Thanks a lot for the reviewer's approval.
This manuscript is a resubmission of an earlier submission. The following is a list of the peer review reports and author responses from that submission.
Round 1
Reviewer 1 Report
Yang Yang et al investigate the role of hepassocin (HPS) on ER stress and its involvement in establishment/worsening of liver steatosis. They first assess mechanism of HPS induction and establish a feedback with ER stress phenomenon. Then they evaluate its function in ameliorating liver disfunction in different model leading to steatosis. Moreover they associate HPS-mediated phenotype to calcium homeostasis. Although the topis is interesting and would provide valuable information to a board spectrum of fields, many clarifications are needed and a valid molecular mechanism needs to be shown in support of authors conclusions. Major revision is needed.
1. The first two phrases of the abstract appear confusing and contrasting. Please provide clarification.
2. It would be very important to clarify the difference between tunicamycin model and MCD diet in studying steatosis.
3. It is not clear if HPS is also expressed in other cells systems, or if it acts in trans from hepatocytes.
4. Authors should clarify the novelty of their study in comparisons to the study in ref 16
5. It would be important to provide a valid rationale for inhibiting XBP1.
6. Please clarify, providing references on molecular mechanisms, why is PERK important during ER stress.
7. It would be important to add a control for authors recombinant protein reconstitution assays. Authors could try and overexpress it, but also try to reconstitute with other “ER inert” proteins and evaluate specific effect.
8. Please spell TG the first time it is cited in the main results regardless of the methods. Moreover, authors should clarify that TG is not “only” ER stressor for hepatocytes and explain TG direct and indirect roles on ER.
9. English grammar error need to be corrected.
10. In fig1A please indicate what gene is being assessed.
11. Authors never provide explanation for the finding that there an increase in intracellular calcium in normal ER stress condition.
12. It is not clear what figure/panel authors are referring to when they mention : “The basic transcriptional activation of the UPR genes was compared between HPS KO liver and WT liver, indicating normal UPR in liver of HPS KO mice.”
13. Provide a conclusion for paragraph 3.6
14. First part of the discussion is major typos derived from previous submissions, please delete.
15. It appears that the findings about calcium are the only somewhat mechanism the authors possess for the effect of HPS. However this is not sufficient, authors could use calcium ionophores and evaluate proteins related to calcium efflux in the KO and therefore trying to strengthen the mechanism. Besides calcium, a valid molecular mechanism of action needs to be provided. Authors could also look at potassium and assess its role in the effect mediated by HPS.
16. Authors should clarify how HPS tunes the transcription of the mentioned genes. Is the effect direct at all or just via feedback on ER stress? Moreover authors should explain better how the feedback mechanism works.
17. It would be very important that authors perform 2 way anova statistical analyses when 2 groups are assessed.
18. In the experiments of reconstitution with recombinant protein authors could use other ER stress inhibitors to strengthen their findings in possibly reporting less strong effect of HPS. This is to prove unequivocally that the effect of HPS on the parameters assessed is given by alleviation of ER stress.
19. It is not clear why baseline TM treatment does not result in calcium increase but TG is needed.
20. In fig 4C authors should include the fasting 0h (meaning fed). In fig 4D-E as well “fed” needs to be shown. Moreover statistical analysis between comparison 0-24h needs to be shown in Fig4C.
21. Authors should change their bar graph into scatter plots with individual points shown.
22. Authors should assess serum TG levels in WT and KO under TM.
23. It would be important to provide quantifications/scorings and corresponding graphs of fig 6B, 7C and 8E.
24. It is not clear if cytokines in fig 6D are assessed in serum or total liver.
25. Control mice (no TM) should be included in qPCR in fig 6F.
26. Authors should provide info regarding the possibility/fact that other organs are affected by TM injection. If so, do authors see big differences in the KO in these organs too?
27. It would be important to assess potential amelioration of phenotype if authors tried to inject HPS after TM. This will help shed light on mechanisms of action and clarify questions of protection vs amelioration etc..
28. Specify on the panel 7B itself that it refers to KO only.
29. Please add control mice in fig 8A-B-C-D.
30. It is not entirely clear why authors do not decide to use PBA in other experiments than just fig8. (see point n18)
Reviewer 2 Report
In this study, the authors, aimed to investigate the function of Hepassocin (HPS) in physiological and drug-induced hepatic endoplasmic reticulum (ER) stress in vitro and in vivo. They found that HPS acts as a negative feedback regulator for hepatic ER stress. They also showed that inhibition of ER stress may be an important mechanism for HPS attenuation NASH development in mice. I have no comments as the study is of interest and both design and results are clearly presented and discussed. The study has potential clinical impact as the results would suggest that HPS or targeted its signaling pathway might have therapeutic potential for the ER stress related liver injure.
To improve the clinical impact of the manuscript I only would suggest discussing the potential therapeutic application and future research strategies.